# Environmental factors and occurrence of horseshoe crabs in the northcentral Gulf of Mexico

**Maurice G. Estes, Jr.**[1]*, **Ruth H. Carmichael**[2☉], **Xiongwen Chen**[3☉], **Sean C. Carter**[4]

**1** Atmospheric and Earth Sciences Department, University of Alabama in Huntsville, Huntsville, Alabama, United States of America, **2** Department of Marine Sciences, University of South Alabama, Dauphin Island, Alabama, United States of America, **3** Biological and Environmental Sciences Department, Alabama Agricultural and Mechanical University, Normal, Alabama, United States of America, **4** Department of Ecosystems and Conservation, University of Montana, Missoula, Montana, United States of America

☉ These authors contributed equally to this work.
* maury.estes@nsstc.uah.edu

**Data Availability Statement:** The data is available publicly via the Dauphin Island Sea Lab Data Management Center and is at this link: Carmichael, Ruth, and Maurice Estes (2016). American Horseshoe Crab Abundance in the Northern Central

## Abstract

This study provides regional-scale data on drivers of horseshoe crab (*Limulus polyphemus*) presence along the northcentral Gulf of Mexico coast and has implications for understanding habitat suitability for sparse horseshoe crab populations of conservation concern worldwide. To collect baseline data on the relationship between environmental factors and presence of horseshoe crabs, we surveyed four sites from the Fort Morgan peninsula of Mobile Bay, Alabama (AL) to Horn Island, Mississippi (MS). We documented number, size and sex of live animals, molts, and carcasses as metrics of horseshoe crab presence and demographics for two years. Data were compared to *in situ* and remotely sensed environmental attributes to assess environmental drivers of occurrence during the time of study. Overall, greater evidence of horseshoe crab presence was found at western sites (Petit Bois and Horn Islands) compared to eastern sites (Dauphin Island, Fort Morgan peninsula), mediated by a combination of distance from areas of high freshwater discharge and interannual variation in weather. Higher sex ratios also were found associated with higher occurrence, west of Mobile Bay. Land cover, particularly Bare Land and Estuarine Emergent Wetland classes that are common to western sites, was most predictive of live animal and to some extent carcass occurrence. Our findings suggest that small-scale variation in habitat quality can affect occurrence of horseshoe crabs in sparse populations where density is not a limiting factor. Data from molts and carcasses were informative to supplement live animal data and may be useful to enhance ecological assessment and support conservation and management in regions with sparse populations.

Gulf of Mexico in 2012-2013. Data Management
Center, Dauphin Island Sea Lab, Alabama, USA.
Available at: https://catalog.data.gov/dataset/
american-horseshoe-crab-abundance-in-the-
northern-central-gulf-of-mexico-from-2012-05-21-
to-201. Last Accessed: 11 June 2020.

**Funding:** The project was supported by the
National Science Foundation Research Experiences
for Undergraduates program at the Dauphin Island
Sea Lab and Xiongwen Chen was partially
supported by USDA Mc-Stennis project (1008643).
The funders had no role in study design, data
collection and analysis, decision to publish or
preparation of the manuscript.

**Competing interests:** The authors have declared
that no competing interests exist.

## Introduction

The northcentral Gulf of Mexico (GOM) represents the westernmost habitat in the U.S. for the
American horseshoe crab (*Limulus polyphemus*), which is categorized as a "vulnerable" species
by the International Union for Conservation of Nature (IUCN) red list [1–3]. Previous
research in the northcentral GOM indicates sparse presence of *L. polyphemus* among the
sandy beaches on the lee shore of the barrier island system that extends from Louisiana to the
Fort Morgan peninsula in Alabama [2, 4]. *L. polyphemus* is also found in the Yucatan penin-
sula of Mexico, which is the southernmost population of the species [5]. Numbers of horseshoe
crabs are reportedly higher to the east along the Florida panhandle [6], with the greatest num-
bers of horseshoe crabs in the U.S. occurring on the mid-Atlantic coast [7–9]. Accordingly, the
mid-Atlantic coast is the most well-studied and characterized habitat for horseshoe crabs in
the world. Detailed data on abundance, distribution, and habitat use among horseshoe crabs
in the northcentral GOM and similar sparsely populated areas are limited by a lack of study in
these outlying habitats [4, 9]. These habitats, however, may be increasingly important as habi-
tat degradation and climate change alter habitat suitability for horseshoe crabs and their prey
species elsewhere [10].

Abiotic factors influence the use of habitat and determine horseshoe crab range by charac-
terizing the physical and chemical boundaries of survival as they do any species. Major abiotic
factors that affect the global range of horseshoe crabs include temperature, tidal excursion and
frequency, geologic setting, and benthic currents [10, 11]. Tidal magnitudes determine the
timing of spawning during moon cycles, with animals tending to swim with tidal currents to
the shore and spawn at highest densities in regions with a wider tidal range [12, 13]. Limited
tidal ranges also affect horseshoe crab movement and habitat use, which may explain why
horseshoe crabs are not found west of Louisiana in the northwestern GOM [14]. Other vari-
ables that may contribute to local horseshoe crab habitat suitability include wave height, wind
speed, wind direction, salinity, dissolved oxygen, water quality, beach slope, near shore
bathymetry, sediment grain size, and drainage capacity of sediments [2–3, 15–18]. In the
northcentral GOM, freshwater outflow from the Mobile Bay and Mississippi River watersheds,
which are among the largest in the U.S., discharge east and west of the barrier island system
that comprises the known habitat for horseshoe crabs. Freshwater discharges influence estua-
rine salinity, which at levels of 8–10 parts per thousand (ppt) and below are known determi-
nates for horseshoe crab habitat suitability [19], suggesting freshwater discharge and
associated variation in salinity may affect horseshoe crab occurrence along the northcentral
GOM coast. Climate, which affects many of these same abiotic factors, has potential to further
alter habitat and horseshoe crab occurrence through time [10, 20]. By affecting distribution
and range boundaries, interactions between freshwater discharge and other environmental
factors may contribute to the low density of horseshoe crabs in the northcentral GOM region
and have potential to make these sparse populations susceptible to local extinctions.

Horseshoe crab habitat also may be affected by direct anthropogenic modification of land-
scape and waters. Urbanization that alters shorelines can eliminate spawning and nursery or
foraging habitat by reducing the area of sandy beach [21]. Previous study suggests horseshoe
crabs may be particularly sensitive to and slow to recover from changes in habitat geomorphol-
ogy [22]. Increased bare land and impervious surfaces [23], associated runoff and changes to
water chemistry, in turn, affect sediment grain size and moisture, which can further affect
horseshoe crab spawning behavior and egg development [16, 22, 24]. Urban pollutants such as
heavy metals and other contaminants that impair water quality may slow development and
cause deformities of juvenile horseshoe crabs [25, 26]. Hence, changes to the quality or spatial

configuration of habitat can affect the occurrence of horseshoe crabs by affecting distribution or survival at life stages from eggs and juveniles to adult animals [10, 19, 27].

Remotely sensed data can supplement, and in some cases replace, *in situ* data needed to relate horseshoe crab occurrence to habitat characteristics and assess suitability [28]. Daily availability of satellite data can reduce spatial and temporal data gaps, particularly for populations in poorly studied regions, and reduce the need and associated expense of field data collection [28]. This approach is growing in popularity and is arguably essential to the future of conservation ecology [29, 30], given the high temporal availability and global coverage of satellite data for terrestrial and aquatic habitats. Remotely sensed data have potential to enhance understanding of the functional relationships between shoreline characteristics and horseshoe crab occurrence. In areas with limited data, combining remotely sensed with *in situ* data will better support establishing habitat baselines and documenting change through time in the future.

While live animals are preferred for demographic and occurrence studies, surveying live horseshoe crabs can be problematic, particularly in areas of low population density. First, the different age classes of horseshoe crabs are difficult to study with a single survey type or plan. The different age classes of live horseshoe crabs occupy different habitats throughout life, with young staying close to nursery grounds for at least the first two years of life and most adults thought to migrate from shelf or offshore waters seasonally [31] Adults primarily aggregate nearshore and come to beaches to spawn in the spring in the U.S. Second, horseshoe crabs often occur in areas that are not easy to access, with sloping shorelines or in areas with turbid waters that make study difficult [4]. Accordingly, most previous demographic studies in the U.S. were done either in shallow enclosed waters with high clarity or with trawls or spawning surveys [32–34]. Recent improvements to trawl survey methods better capture small animals but still miss the youngest size classes that are too nearshore or too small to be captured [35]. Similarly, spawning surveys are limited largely to adults and only those that come to shore to spawn. For this reason, there are few whole population demographic studies of horseshoe crabs anywhere in their range 32–33. Finally, typical survey methods are most useful in densely populated areas like the mid-Atlantic, but may be too inefficient and impractical in large and less densely populated areas [33, 35]. Hence, there is need for alternative approaches to better study sparse populations like those in the northcentral GOM and elsewhere in the world outside the U.S. Atlantic coast.

This project is a first step to provide baseline data on the relationship between environmental factors and the presence of sparse horseshoe crab populations along the northcentral GOM coast. This study is unique in combining data from live animals, molts, and carcasses as three metrics of horseshoe crab presence. Horseshoe crab data (numbers, size, sex) were compared to *in situ* attributes (salinity, water temperature, dissolved oxygen, water clarity, bathymetry) and remotely sensed environmental data (daily maximum wind speed, daily wind direction, daily total precipitation, land cover land use) to assess environmental drivers of occurrence. We hypothesized that the occurrence of horseshoe crabs among sites would increase with increasing distance from the Mobile Bay estuary, the 3rd largest freshwater discharge among rivers with watersheds in the U.S. [36], and it would be influenced by shoreline land cover land use (LCLU) and near shore water conditions. We found that interannual variation in weather and within site variation in LCLU may be as important to determining horseshoe crab presence as environmental variation among sites alone. This study has implications for understanding horseshoe crab occurrence and habitat suitability for any sparse population, including data deficient species in similar habitats in Southeast Asia [37, 38].

## Study area

This study was conducted at four sites along the northcentral GOM coast including Horn Island, MS; Petit Bois Island, MS; Dauphin Island, AL; and Fort Morgan peninsula, AL (Fig 1). The three islands are part of the barrier island chain that separates Mississippi Sound from the GOM to the south. Petit Bois Island and Horn Island are part of the Gulf Islands National Seashore (GINS) that is managed by the National Parks Service (NPS). The Fort Morgan peninsula also is part of the Bon Secour National Wildlife Refuge in Alabama. The Alabama-Mississippi coastal region is characterized by estuarine and palustrine wetlands, forests, agriculture and developed land. The geomorphology of the region has been shaped by fluvial processes and sea level rise (SLR) during the last 7,000 years [39], with topography dominated by coastal plain flat lands and gentle slopes at an elevation near sea level. The combination of SLR and loss of sediment from rivers discharging in the Mississippi Sound estuary has reduced the size and stability of the barrier islands [40], and the intensity and frequency of tropical storms has exacerbated coastline erosion [41]. The major marine currents for the GOM are the Gulf Stream and associated loop current, south of the study area. Diel micro-tides (range of ~0.5 m) and wind driven longshore currents transport material and influence the water movement, predominately from east to west in the study area [42]. Average wave heights are 0.6 and 0.4 m respectively for the winter and summer seasons, and wave periods average 3–4 seconds year-round [40]. The study

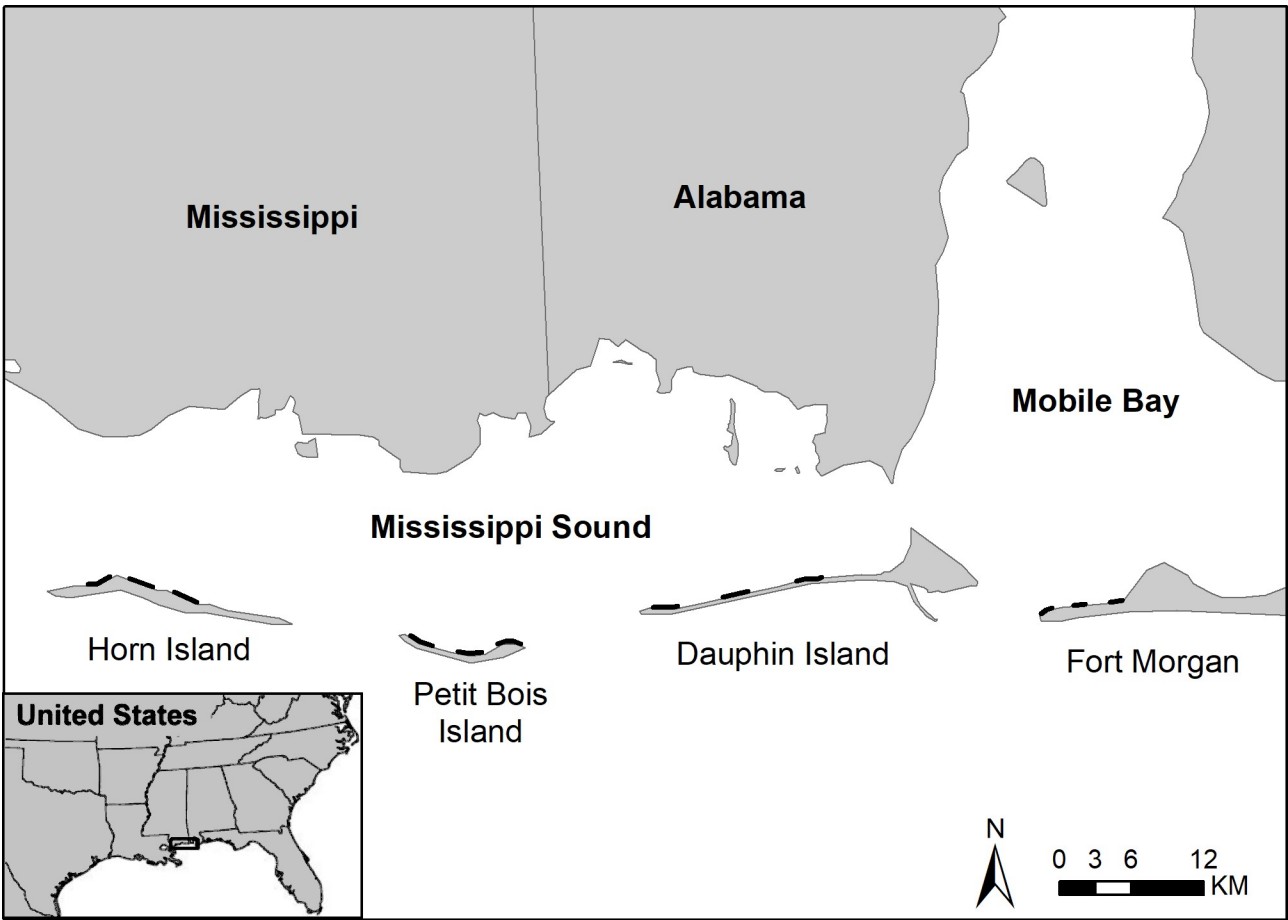

**Fig 1.** Study area within the northcentral GOM, USA (left), including barrier islands (right) and locations of the four shoreline transects (black lines) for horseshoe crab surveys and in situ environmental data collection during 2012 and 2013.

area is in the Koppen climate classification, having moist subtropical weather with mild winters and warm humid summers. Precipitation is typically high year-round in the region, averaging 1000–1800 mm annually. Freshwater discharge into the northcentral GOM varies seasonally, typically with higher discharge in winter and spring than in summer and fall [43]. The average annual temperature in the study area is 20.6˚C (ranging ~5˚C to 30˚C).

## Materials and methods

### Horseshoe crab sampling

To define horseshoe crab occurrence, the numbers of live horseshoe crabs, molts (exoskeletons shed to accommodate growth), and carcasses were recorded during field surveys on the leeward side of four sites, including the Fort Morgan Peninsula, Dauphin Island, Petit Bois Island, and Horn Island (Fig 1). Surveys were conducted from May to August at biweekly and monthly intervals (coincidental with full and new moon phases) in 2012 and 2013, respectively, plus additional monthly surveys in September and October of 2012, with some variation in schedule due to weather conditions. Thirty-two surveys were completed in 2012 (7 each on Dauphin Island and Fort Morgan Peninsula and 9 each on Petit Bois Island and Horn Island) and 15 surveys were completed in 2013 (4 each on Fort Morgan peninsula, Petit Bois Island and Horn Island, and 3 on Dauphin Island), ensuring that a sufficient number of surveys were completed under different environmental conditions to relate horseshoe crab counts to environmental co-variates at each site. Surveys were conducted along three transects at each site during the daytime within ± two hours of high tide (Fig 1). The transects were 2 km in length for each of the barrier island sites and ranging from 0.5 to approximately 1 km for the 3 transects on Fort Morgan Peninsula due to limited beach access as a result of eroding shorelines and urban development. No nighttime surveys were performed due to safety and logistics related to accessing remote locations on barrier islands.

The protocol for field surveys was to walk the water line and search for live crabs, carcasses and molts on the beach area and in the water as far as visibility allowed from the water line (~ 5 m total width). Each specimen found was classified as live, molt or carcass, GPS coordinates were noted, and paired (in amplexus) status was recorded for live animals. Prosomal width (PW) was measured to the nearest mm using a clear plastic ruler placed across the widest part of the carapace, about 2.5 cm above the animals' hinge for subadults and adults (S1 Fig). Calipers were used for smaller, juvenile molts to determine prosomal width to 0.1 mm. Live animals were counted, prosomal width measured, and sex determined from visual examination and then released. All work was conducted under a permit from the National Park Service, Gulf Islands National Seashore (United States Department of the Interior, National Park Service, Gulf Islands National Seashore Study# GUIS-00191, Permit# GUIS-2012-SCI-0031).

Sex classification of male, female or unknown was made based on the presence of genital slits (female) or pores (male), modified pedipalps (male), and/or height of the prosomal arch (higher in males). All molts and carcasses found were removed from the site by either collection or marking with paint and discarding in the dunes or marsh areas off the primary beach to avoid recounting. Live animals were released moving away from the direction of the survey to avoid recounting the same animals during each survey; the sparse numbers of live animals made it easy to avoid recounting, and previous study has shown recovery rates are low for horseshoe crabs in similarly structured mark-recapture studies [44].

### Site fidelity of molts

To test the likelihood that molts found on local beaches originated from and thus represent local populations of live animals, a molt transport experiment was conducted in the summer

of 2015 on Dauphin and Petit Bois Islands. Molts were tagged with 2.5 cm x 1.5 cm flexible rectangular plastic tags obtained from the U.S. Fish and Wildlife Service. Tags were affixed with waterproof epoxy to avoid damaging the shell. Tagged molts (n = 65 per site; DI: 99.7 ± 2.4 mm, PB: 99.5 ± 2.7 mm PW) were placed on a grid at 100 m intervals along a 2 km transect, 2 m, 50 m and 100 m from shore on the north side of each island during June. The experiment was repeated on Dauphin Island in July (n = 65; 96.8 ± 2.4 mm PW). Transects for the molt transport experiment were coincident with the westernmost and easternmost portions of transects shown in Fig 1 for Dauphin Island and Petit Bois Island, respectively.

## Environmental attributes

Concurrent with the horseshoe crab field surveys, *in situ* environmental attributes were measured in the water column along nearshore transects, parallel to each survey line, approximately 0.2 km from shore. Samples were taken at the starting point of each survey and every 0.5 km along the transect using a YSI handheld meter to measure water temperature, salinity, and dissolved oxygen (Fig 1). Measurements were taken at 0.3 m from the bottom to capture conditions relevant to horseshoe crab movements. Total water depth and secchi depth were measured to define water clarity. For reference to the environmental conditions during the field surveys, climatic mean values for Dauphin Island were compiled for the months of June, July, and August based on a statistical analysis of historical weather reports and model reconstructions from 1980 to 2016 [45].

For bathymetry, digital elevation model (DEM) data for the northcentral GOM coast were acquired from the National Oceanographic and Atmospheric Administration's (NOAA) Geophysical Data Center (NGDC). The DEM is a grid divided into discrete cells of identical size that are tiled to spatially fill the grid region without gaps. The elevation value of each cell was derived from the values of individual elevation point data that lay within that cell's spatial coverage of 1 km or from the values of neighboring cells. Each cell elevation value (meters) thus defines an average or approximate elevation for the entire cell computed with the NOAA VDatum DEM Project (NOAA's National Geodetic Survey, Office of Coast Survey and Center for Operational Oceanographic Products and Services, Silver Spring, MD, USA). For the NOAA VDatum DEM Project, the defined horizontal datum is North American Datum of 1983 (NAD83), in decimal degrees of latitude and longitude, and defined vertical datum of mean high water (MHW) in meters. Bathymetry values (ocean floor elevations) for each 1 km grid cell were retrieved for a location within a few meters of the shore and a value 1 km offshore due north of the centroid of the grid cell and computing the slope for a straight line between these points. The nearshore value and 1 km distance due north were obtained using ArcGIS (Environmental Systems Research Institute, Inc., Redlands, CA, USA).

Discharge data were acquired from the U.S. Geological Survey for the Mobile River at River Mile 31.0 at Bucks, AL for the period of May through September in both 2012 and 2013. Discharge data were converted from 15-minute time steps to daily mean flows for each year to facilitate evaluation of the discharge effects on environmental conditions and horseshoe crab occurrence.

We used remotely sensed products from the North American Land Data Assimilation System (NLDAS) [46–48] to evaluate the importance of local meteorology, including daily maximum wind speed (m/s), wind direction at the time of maximum wind speed (degrees from north, clockwise) and daily total precipitation (mm) on horseshoe crab occurrence. NLDAS grid cells extend offshore just north of the barrier islands and Fort Morgan peninsula study sites. The 12 km grid cell nearest each transect on the day of the survey was used to determine daily attribute values.

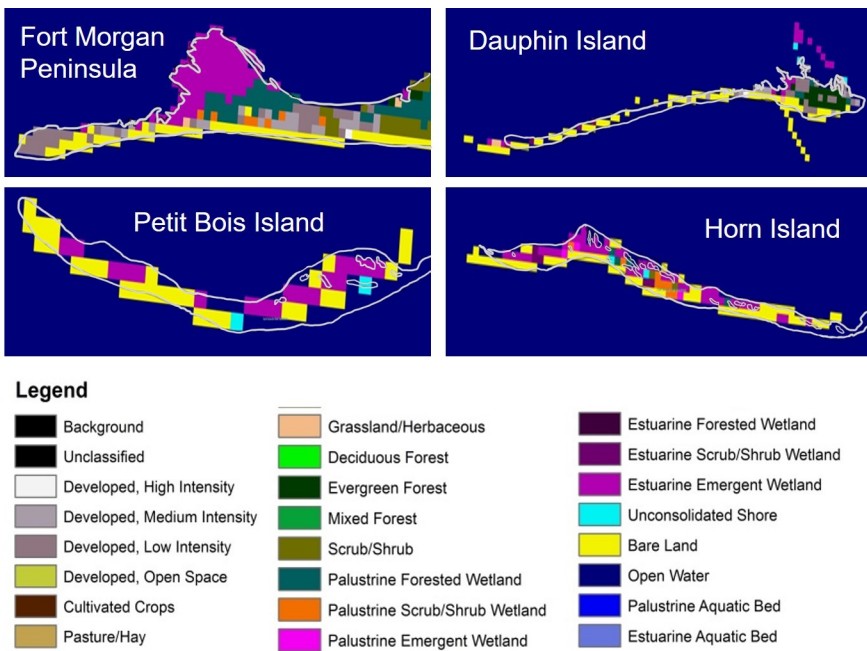

**Fig 2. National Oceanic and Atmospheric Administration coastal change analysis program land cover land use classes resampled from the native resolution of 30 m to an area with a spatial scale of 300 m to align with horseshoe crab shoreline habitat.**

LCLU data were derived from Landsat thematic Mapper imagery and the LCLU classes [49] obtained using the Coastal Change Analysis Program (C-CAP) as a layer in ArcGIS 10.2.1 with a base map of the sites. Resampling at various scales was evaluated to identify the most suitable scale for determining a dominant class to link with horseshoe crab presence data (Fig 2). The dominant class was intended to represent an area of environmental space that will permit horseshoe crabs to forage for food and reproduce [50] and to constitute an area large enough that correlations with numbers of live horseshoe crabs, molts or carcasses would not be random.

The dominant class was selected for the shoreline area (1 km pixel length x width of 5 m) for each 1 km pixel within each transect and site by visual evaluation in ArcGIS. The connectivity of dominant classes within each site was not considered because access to the shoreline was readily available for horseshoe crabs along all shorelines surveyed. Proportions of selected LCLU classes were computed by measuring the length of the class for each site using the Arc GIS 10.2.1 measuring tool, computing area of each class and dividing by the total area of the site. GPS coordinates for horseshoe crab data types were used to spatially join in ArcGIS with C-CAP LCLU classes to determine the number of live horseshoe crabs, molts and carcasses per LCLU class.

## Data analyses

The Microsoft Excel (14.7.1) data analysis tool was used to compute descriptive statistics for the study areas in 2012 and 2013. Differences in numbers and sizes of animals were compared through time using linear regression to determine significant relationships. Because more surveys were conducted in 2012 than 2013, comparisons between 2012 and 2013 data used only the coincident sampling dates between years to compare the same time periods. Arithmetic means were computed for all the horseshoe crab data types and environmental data (except

wind direction) for survey years 2012, 2013 and for combined 2012–2013 data for the study area. Sex ratios were computed by site for all data that allowed for a sex determination. Mean wind direction for each site and year was calculated using the wind direction (angle in degrees) measured during each survey. The sin and cosines for each wind direction value coinciding with surveys was computed, and values were summed and divided by the total number of surveys for that site. The ratio of the summed sines ($\bar{s}$) to summed cosines ($\bar{c}$) was computed and the arctangent of the ratio represented the mean wind direction with these rules applied as appropriate [51]:

a. If $\bar{c} < 0$, + 180 degrees

b. If $\bar{s} < 0$, $\bar{c} > 0$, + 360 degrees

General linear modeling (GLM) in the Standard Statistical Package for the Social Sciences (SPSS V.21) was used to determine if site differences among data types were significant using the multivariate Wilks' lambda and Tukey's post-hoc tests. Model design was a GLM, full factorial sum of squares III with live animals, molts, and carcasses as dependent variables and site or year as the fixed factor with no covariates. The GLM model design for testing environmental influences was also a full factorial sum of squares III with dependent variables water temperature, salinity, water clarity, dissolved oxygen, wind speed, wind direction, and precipitation, fixed factors site and year, and the survey day of year was a covariate. The Least Significant Difference method was used to determine significance among comparisons of the variables to each site and year with an alpha value of 0.05 to determine significance. Due to uneven sample sizes and variances, the Tamhane post-hoc test was used to evaluate mean sample size differences of all data types collected among the four sites. The relationship between environmental variables and counts for live animals, molts and carcasses was evaluated with the SPSS V.21 curvilinear analysis tool that allows for goodness of fit to be determined using linear, quadratic, cubic and other models.

For LCLU data, GLM was used to determine if the difference in live animals, molts or carcasses and LCLU class was significant on a site basis and for the total study area. Molts were combined with live animals because molts are a proxy for juvenile horseshoe crabs that could be affected by LCLU change. The Wilks Lambda test was used to determine the significance of multivariate effects for horseshoe crab data types compared to LCLU, LCLU + site and site. Univariate effects were determined with the data type (live, molt, carcass) as the dependent variable and the LCLU classes and site as single factors and the combined effect of LCLU + site as factors using a sum of squares III main effects model design. Correlations between live horseshoe crab occurrence and LCLU classes were computed and evaluated using the Kappa statistic designed to compare the agreement against that which might be expected by chance (Microsoft Excel 14.7.1). The Kappa statistic takes into account both observed agreement and expected agreement, thereby adjusting for chance agreements given the percentages of suitable shoreline classes among the study sites. For the Kappa Statistic, values of 0.61 to 0.80 indicate substantial agreement of relationships and 0.81 to 0.99 very strong agreement [52, 53].

## Results

### Spatial and temporal distribution of horseshoe crabs

A total of 52 live adult horseshoe crabs, 476 molts and 327 carcasses were found during the study (Table 1). Significant differences in the numbers of live horseshoe crabs, molts and carcasses were found among the four sites (GLM: Wilks Lambda Test: $F_9 = 4.11$, $p < 0.001$; Fig 3). Overall, greater evidence of horseshoe crab presence was found at western sites (Petit Bois

**Table 1. Environmental attributes sampled at Fort Morgan (FM), Dauphin Island (DI), Petit Bois Island (PB), Horn Island (HI) (listed from east to west) during the years of 2012 and 2013.**

| 2012 | L | M | C | Temp. (°C) | Salinity (ppt) | DO (mg L$^{-1}$) | WC (m) | Precip. (mm d$^{-1}$) | WD (°) | WS (m s$^{-1}$) |
|---|---|---|---|---|---|---|---|---|---|---|
| FM | 15 | 0 | 2 | 28.7 ± 0.8 | 27.6 ± 1.4 | 6.1 ± 0.3 | 0.8 ± 0.1 | 0.3 ± 0.1 | 192 ± 13 | 5.2 ± 0.9 |
| DI | 0 | 0 | 43 | 28.7 ± 0.4 | 29.1 ± 0.9 | 6.0 ± 0.3 | 1.1 ± 0.1 | 1.0 ± 0.7 | 196 ± 13 | 4.8 ± 0.7 |
| PBI | 27 | 41 | 100 | 28.3 ± 0.6 | 29.7 ± 1.0 | 6.0 ± 0.3 | 1.4 ± 0.1 | 1.1 ± 0.5 | 214 ± 21 | 5.1 ± 0.5 |
| HI | 9 | 39 | 137 | 28.0 ± 0.7 | 27.8 ± 1.1 | 6.1 ± 0.3 | 1.3 ± 0.1 | 1.7 ± 1.0 | 203 ± 19 | 4.9 ± 0.4 |
| 2013 | | | | | | | | | | |
| FM | 1 | 0 | 1 | 27.1 ± 1.0 | 18.3 ± 3.3 | 4.7 ± 0.6 | 0.6 ± 0.1 | 3.1 ± 1.5 | 152 ± 39 | 6.4 ± 0.8 |
| DI | 0 | 0 | 12 | 28.0 ± 0.5 | 21.1 ± 2.0 | 5.5 ± 0.5 | 0.9 ± 0.1 | 5.8 ± 5.0 | 120 ± 36 | 6.0 ± 0.7 |
| PBI | 0 | 394 | 19 | 28.1 ± 0.4 | 24.1 ± 2.5 | 5.5 ± 0.5 | 1.3 ± 0.1 | 1.2 ± 0.9 | 184 ± 21 | 5.9± 0.6 |
| HI | 0 | 2 | 13 | 27.6 ± 0.4 | 21.6 ± 2.9 | 5.5 ± 0.4 | 0.9 ± 0.1 | 1.2 ± 0.9 | 193 ± 10 | 5.8 ± 0.3 |

Temp = water temperature, DO = dissolved oxygen, WC = water clarity, Precip = precipitation, WD = wind direction, WS = wind speed.

Island and Horn Island) compared to eastern sites (Dauphin Island and Fort Morgan peninsula), (GLM: carcasses, $F_3$ = 5.49, p < = 0.01; molts, $F_3$ = 4.81, p = 0.01). The number of live animals found at each site was small and similar among sites, but molts and carcasses were highest on Petit Bois and Horn Islands, which are the greatest distance from the Mobile Bay watershed discharge (GLM: Tukey post hoc, p < 0.03 for all significant comparisons). Molts, which are indicative of growing juveniles, also were only found on Petit Bois and Horn Islands; during 2013, 99% of all juvenile molts were found on Petit Bois Island (Fig 3).

Horseshoe crab occurrence also differed between years, with more evidence of occurrence across all sites during 2012 than 2013 (Table 1, Fig 3). Although year of survey was not related

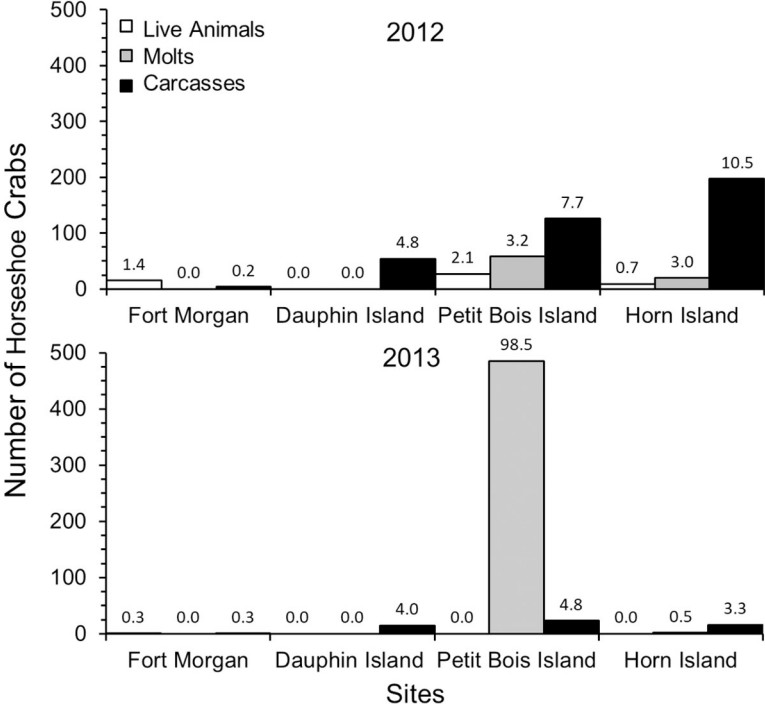

**Fig 3. Number of horseshoe crabs (live animals, molts, carcasses) found at each site during surveys in the years of 2012 and 2013.** The number of horseshoe crabs found per day is noted above each bar to show occurrence normalized to the level of effort among sites.

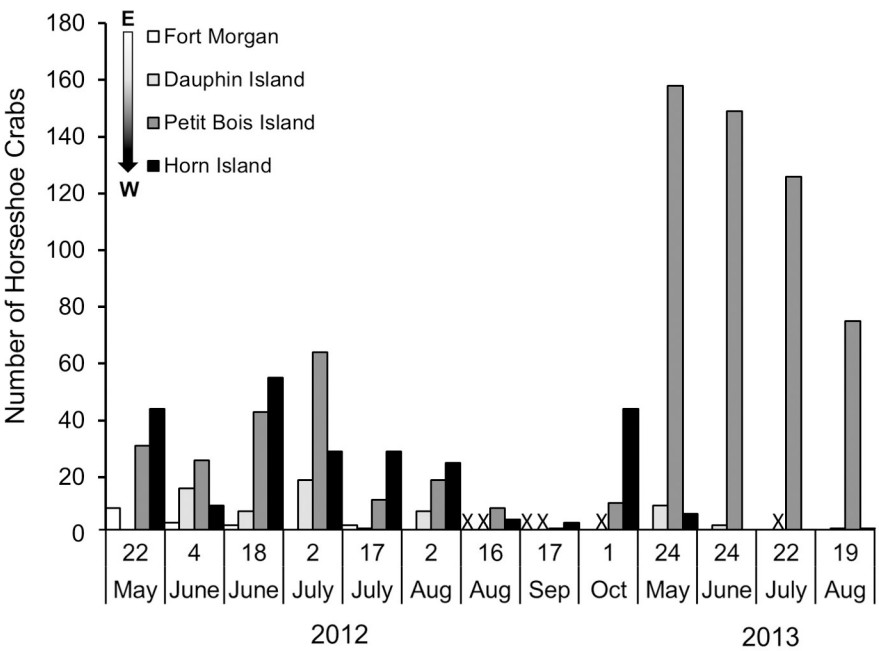

**Fig 4. Number of horseshoe crabs (live, molts, carcasses) found at each site on each survey during the 2012 to 2013 survey period.** The arrow notes the progression from east to west away from the Mobile Bay discharge. No bar indicates 0 horseshoe crabs found, and X indicates no survey.

to numbers of live horseshoe crabs, more live animals were found during 2012 surveys, with only one live adult horseshoe crab found in 2013, at the Fort Morgan site. The number of molts was higher in 2013 than 2012 (GLM: Molts, $F_1 = 11.44$, p < 0.00), due to a very high number of molts found on Petit Bois Island, while carcasses were less abundant in 2013 than 2012 (GLM: Carcasses, $F_1 = 8.37$, p < 0.01). Sample variance and standard error among the three data types (live horseshoe crabs, molts, carcasses) were higher for molts than live horseshoe crabs or carcasses during both survey periods (Table 1). These results did not change when data were normalized to differences in effort among sites (Fig 3, inset numbers). Peak numbers of live horseshoe crabs during this study occurred in late May to early July and decreased for the remainder of the summer during both years surveyed (Fig 4). The number of live horseshoe crabs was significantly related to day of the year for the study domain and Petit Bois Island, for the combined 2012–2013 survey period (Fig 5; Live animals: All sites: $F_{reg46} =$

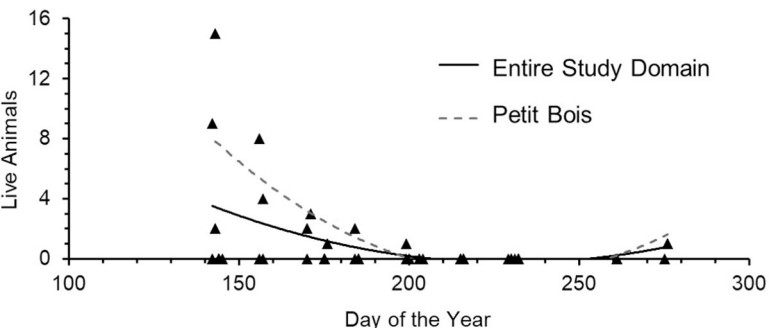

**Fig 5. Relationships between number of live horseshoe crabs found and day of the year during the 2012–2013 survey period for the study domain and Petit Bois Island, MS.**

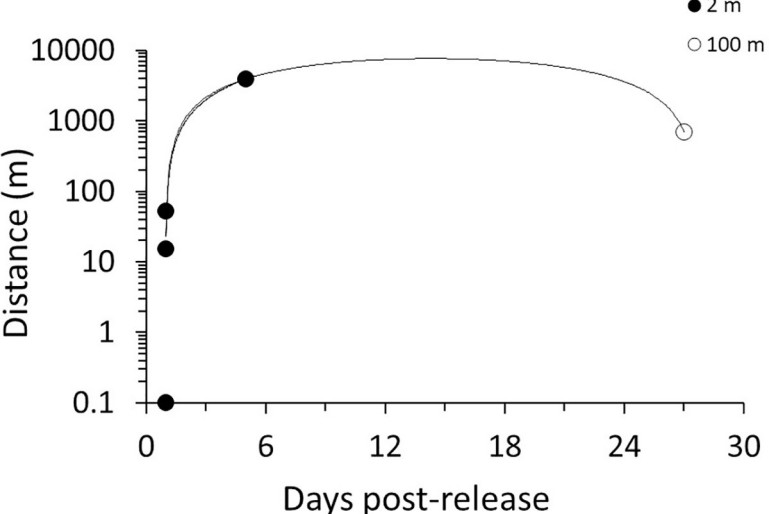

**Fig 6. The distance individual molts traveled alongshore from nearshore (2 m, 100 m) release points during the mark-recapture study.**

6.001, $R^2 = 0.21$, p = 0.01, $y = -0.0005x^2 - 0.226x + 25.70$; Petit Bois: $F_{reg12} = 4.51$, $R^2 = 0.47$, p = 0.04, $y = 0.0012x^2 - 0.543x + 61.26$).

Molt mark-recapture tests confirmed molts likely washed ashore near the time and areas they originated. Recovery of tagged molts was 2.5%, consistent with mark-recapture recovery for other wildlife [5]. All recovered molts were reported within 30 days of original field placement, and as of 2019 no additional tagged molts were recovered (Fig 6). On average, recovered molts traveled 778 m ± 667 m, with a maximum distance of 3.9 km traveled, and all molts were recovered on the same island along which they were originally placed. Most recovered molts (80%) originated 2 m from shore, but one molt washed ashore from placement 100 m offshore.

## Size and sex ratios

Live horseshoe crabs ranged in size from 152 to 280 mm (196 ± 5 mm), carcasses ranged in size from 60 to 300 mm (193 ± 2 mm), and molts ranged from 5 mm to 240 mm (47 ± 2 mm), with animals of 10 to 60 mm comprising the majority of the molt data (Table 1, Fig 7). Among sites, the sizes of live animals and carcasses did not differ, with live animals having a mean size of 194 ± 6 mm for males and 209 ± 9 mm for females, and carcasses having a mean size of 184 ± 2 mm for males and 217 ± 6 mm for females. However, molts were smaller on Petit Bois Island than Horn Island (molts were not found at Dauphin Island and Fort Morgan sites) (GLM; $F_1 = 4.65$, p = 0.049), with the prosomal width of males being 120 ± 8 mm and females 104 ±11 mm. During 2012, the greatest numbers of horseshoe crabs had prosomal widths near 100 mm for molts and 180 mm to 240 mm for carcasses (Fig 7, top). During 2013, data were dominated by molts < 80 mm, with peak numbers at prosomal widths of 16 mm, 36 mm, and 44 mm (Fig 7, middle). Data are shown for each year separately and combined (Fig 7, bottom) to show the alignment of size frequencies between years.

When data for live animals, molts and carcasses were combined for 2012 and 2013, adult male to female sex ratios were 1.1 at Fort Morgan peninsula, 3.3 at Dauphin Island, 2.1 at Petit Bois Island, and 2.3 at Horn Island. For juvenile molts, sex ratios were higher at Horn Island 2.4 than Petit Bois Island 1.2. The majority of the juvenile molts collected at Petit Bois Island were < 10 mm prosomal width and too small to confidently distinguish genital structure.

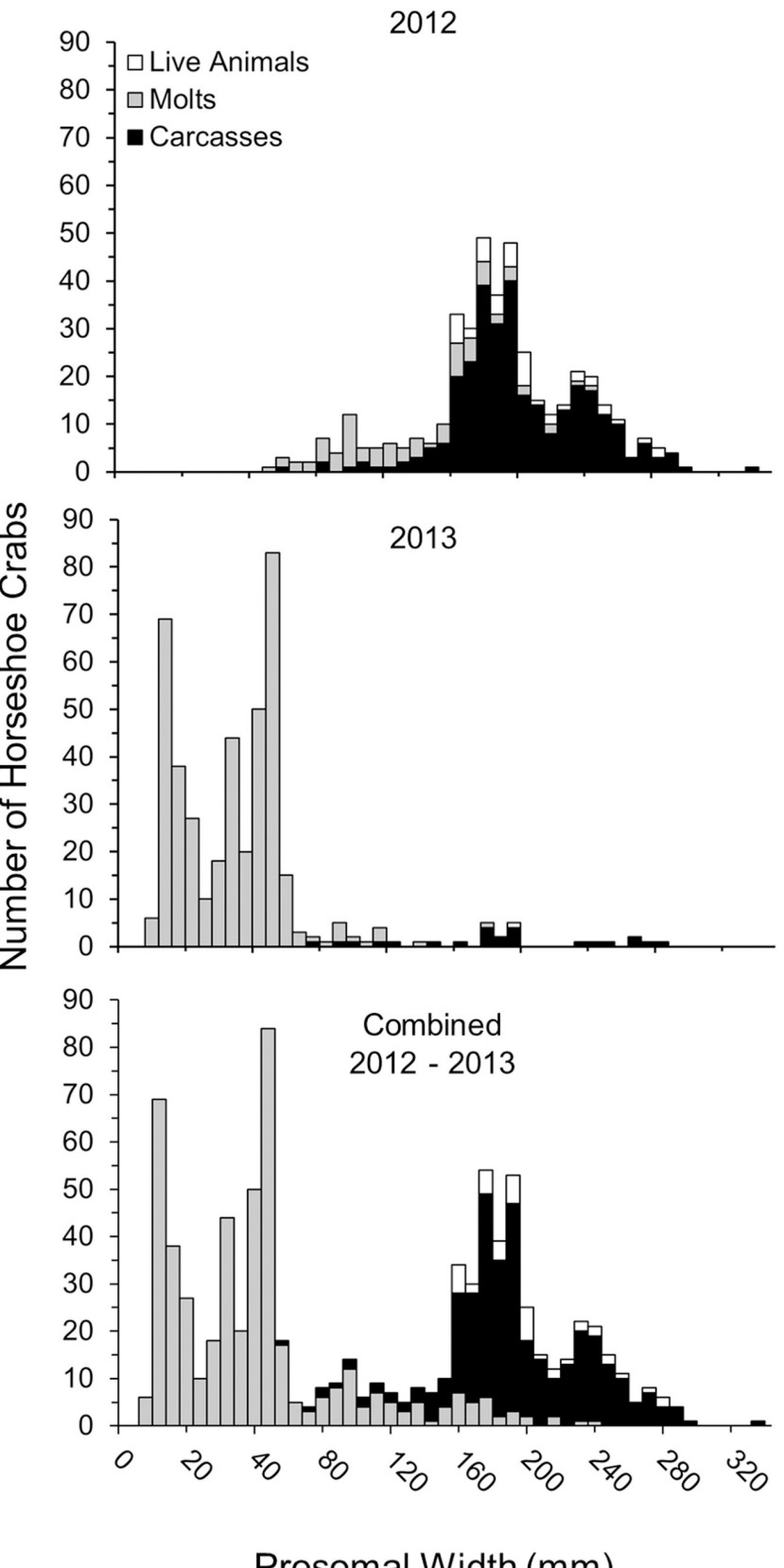

**Fig 7. Size (prosomal width) frequency distribution of horseshoe crabs (live, molts, carcasses) collected during the survey periods in 2012, 2013 and both years combined.**

## Environmental influences

During the survey period, wind direction was consistent with summer climate patterns originating in the west and southwest, however, wind speed was higher than climate means (Tables 1 and 2). Water temperature was consistent with climate means, while precipitation was lower than climate normal overall (Tables 1 and 2).

Environmental conditions were different between 2012 and 2013 for all attributes except water temperature (Table 1). Salinity, DO, and water clarity were lower in 2013 compared to 2012 (GLM: salinity, $F_1 = 30.73$, p < 0.001; DO, $F_1 = 7.12$, p = 0.01; Water Clarity, $F_1 = 10.08$, p < 0.01). Water clarity also was lowest at Fort Morgan and highest at Petit Bois Island, showing an east to west increase in both years (GLM: $F_3 = 14.54$, p < 0.001). Wind speed and precipitation were higher in 2013 than 2012 (GLM: Wind Speed, $F_1 = 4.32$, p = 0.04; Precipitation, $F_1 = 5.40$, p = 0.03). The mean precipitation in 2013 (2.61 ± 1.08) was more than double the mean precipitation in 2012 (1.06 ± 0.33), consistent with lower salinity in 2013. Wind direction was also more from the east than west in 2013 versus 2012 (GLM: $F_1 = 5.63$, p = 0.02).

Discharge data into Mobile Bay from the Mobile River showed higher flows in 2013 compared to 2012 (t-test, p = 0.02). Daily mean flows in 2013 were 23,908 ± 4,256 and ranged from 6,265 to 49,725 cfs (USGS 2018), while daily mean flows in 2012 were 10,503 ± 1,311 and ranged from 5,514 to 20,736 cfs [33] (Fig 8). The largest flow differences between the two years occurred in May.

For the combined 2012–2013 survey period, live horseshoe crab occurrence was associated with wind direction from the southwest at Horn Island ($F_{reg12} = 5.10$, $R^2 = 0.51$, p = 0.03). Carcass occurrence was associated with higher water clarity over the domain and at Horn Island (Domain; $F_{reg46} = 5.10$, $R^2 = 0.25$, p < 0.001; Horn Island; $F_{reg12} = 5.10$, $R^2 = 0.50$, p = 0.03). A prevailing wind direction from the southwest was also related to carcass occurrence at Horn Island (Horn Island; $F_{reg12} = 6.32$, $R^2 = 0.56$, p = 0.02). Significant relationships were not found between other environmental variables and live horseshoe crab or molt occurrence, including bathymetric variables of near shore water depth, water depth 1 km from shore and nearshore slope.

Presence of horseshoe crabs differed with LCLU classification among sites, with the greatest number of live animals, molts and carcasses found adjacent to estuarine and bordering emergent wetland areas (Fig 9). The number of live horseshoe crabs was significantly related to LCLU class and the relationship improved with the addition of site to the model ($R^2$ changed from 0.47 to 0.63). A model design of LCLU + site was also marginally significant for carcasses (Table 3).

**Table 2. Climatic means for June, July and August for selected environmental variables for reference to horseshoe crab survey conditions during field surveys in 2012 and 2013.**

|  | Temp. (ºC) | Precip. (mm) | WD (categorical) | WS (m s⁻¹) |
|---|---|---|---|---|
| June | 27.8 ± 0.5 | 129.5 ± 5.4 | S | 3.7 ± 0.1 |
| July | 29.0 ± 0.2 | 142.2 ± 2.2 | SSW | 3.5 ± 0.04 |
| August | 29.0 ± 0.2 | 129.5 ± 4.7 | WSE | 3.5 ± 0.2 |
| Three Month Mean | 28.9 ± 0.6 | 133.8 ± 4.2 | N/A | 3.6 ± 0.1 |

Temp = water temperature, precip = precipitation, WD = wind direction, WS = wind speed, S = south, SSW = south southwest, and WSE = west to southeast.

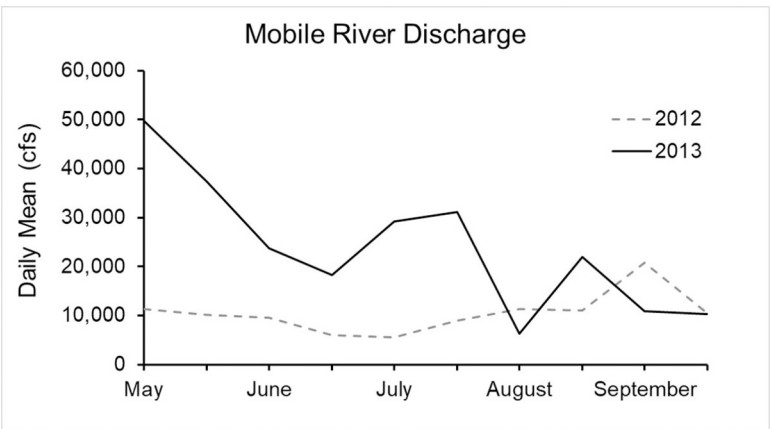

**Fig 8. Daily mean discharge from the Mobile River into Mobile Bay during the 2012 and 2013 survey period.**

On Horn and Petit Bois Islands, live horseshoe crabs and molts were most commonly found along shorelines in the Bare Land and Estuarine Emergent Wetland classes in the western portion of the study area (Fig 9). Live horseshoe crabs were also found in shoreline classes of low and medium development intensity at Fort Morgan to the east. Carcasses were found in the largest number in the Estuarine Emergent Wetland class on both Horn and Petit Bois Islands. Dauphin Island carcasses were most commonly associated with Bare Land and the Estuarine Scrub/Shrub classes. A small number of carcasses were found in the Bare Land class at the Fort Morgan site.

The combination of Bare Land and Estuarine Emergent Wetland (that was associated with the largest number of live horseshoe crabs, molts and carcasses) was most prevalent in the

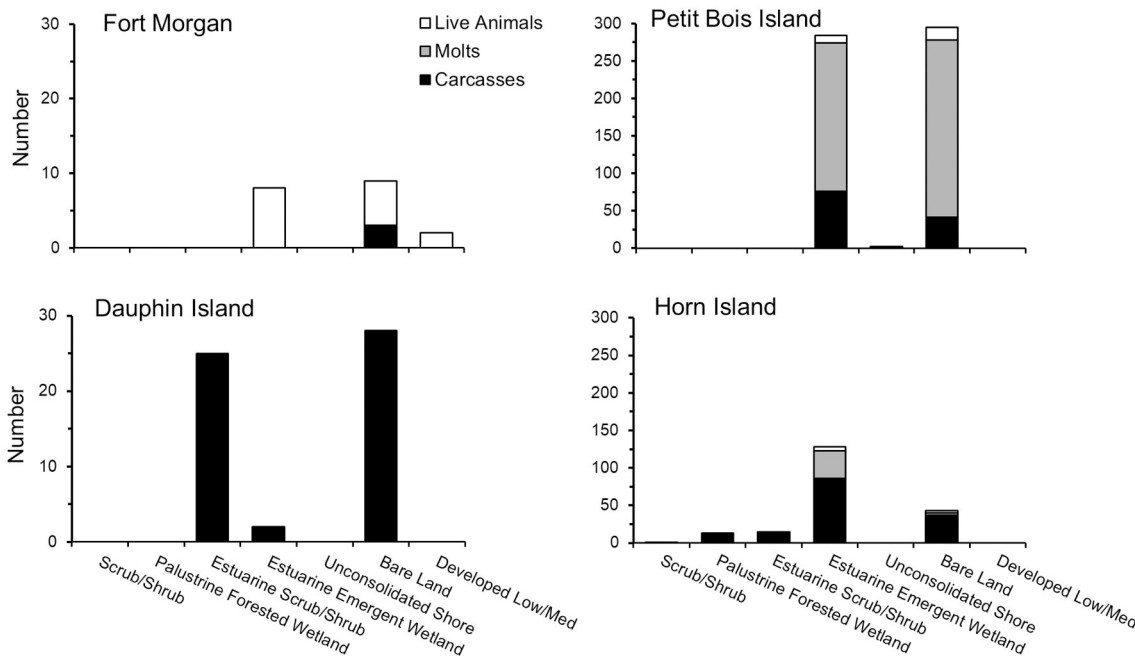

**Fig 9. Numbers of live horseshoe crabs, molts and carcasses found within each land cover land use class at the 4 sites surveyed during 2012 and 2013 along the Mississippi (Horn and Petit Bois Islands) and Alabama (Dauphin Island and Fort Morgan Peninsula) coasts of the northcentral GOM.**

**Table 3. General linear models comparing total numbers of live animals, molts and carcasses to Land Cover Land Use (LCLU) classes and sites discretely and combined (+).**

| Model Design | df | Model (p) | LCLU (p) Site (p) | $R^2$ |
|---|---|---|---|---|
| Live Animals vs LCLU | 6 | 0.039* | | 0.47 |
| Live Animals vs site | 3 | 0.27 | | 0.16 |
| Live Animals vs site + LCLU | 9 | 0.028* | 0.026* 0.13 | 0.63 |
| Molts vs LCLU | 6 | 0.52 | | 0.22 |
| Molts vs site | 3 | 0.08 | | 0.26 |
| Molts vs site + LCLU | 9 | 0.19 | 0.41 0.09 | 0.48 |
| Live Animals + Molts vs LCLU | 6 | 0.22 | | 0.33 |
| Live Animals + Molts vs site | 3 | 0.13 | | 0.22 |
| Live Animals + Molts vs site + LCLU | 9 | 0.10 | 0.17 0.11 | 0.53 |
| Carcasses vs LCLU | 6 | 0.07 | | 0.43 |
| Carcasses vs site | 3 | 0.23 | | 0.18 |
| Carcasses vs site + LCLU | 9 | 0.052 | 0.06 0.16 | 0.59 |

*denotes values significant at alpha $\leq$ 0.05

western portion of the study area at Petit Bois and Horn Islands (Table 4). The Kappa statistic of 0.62 indicated substantial agreement between LCLU classes and occurrence of live horseshoe crabs for the study area or aggregation of all sites (S1 Table).

## Discussion

Greater evidence of horseshoe crab presence in terms of numbers of live animals, molts and carcasses was found at western sites along the northcentral GOM coast, consistent with our hypothesis that horseshoe crab numbers would decrease with proximity to discharge from the Mobile Bay estuary. We also found fewer live animals, molts or carcasses during 2013 when environmental conditions, measured *in situ* and using remotely sensed data, were characterized by higher freshwater flows, lower DO and water clarity, and larger fluctuations of salinity. Salinity values during 2013 at Fort Morgan peninsula at the mouth of Mobile Bay, for example, approached 8 ppt, which is an estimated limiting threshold for horseshoe crab occurrence [13, 54, 55]. Consistent with this view, Petit Bois Island had greater evidence of horseshoe crab occurrence along with higher water clarity and less dynamic salinity and temperature fluctuations compared to other sites. This pattern is meaningful given that the greatest abundance of horseshoe crabs in the northern GOM are located in Florida [9], from which horseshoe crab numbers might be expected to decline westward along the northcentral GOM coast (rather than increasing as we observed) toward the limit of their distribution in Louisiana. Previous studies on Horn Island and West Ship Island (further west) in Mississippi found no difference in relative abundance of spawning adults between these sites [2], suggesting that the area from

**Table 4. Proportion of land cover land use (LCLU) at each site and for the domain (total proportion of LCLU for the total land area) of the four sites.**

| Site | Estuarine Emergent Wetland | Bare Land | Estuarine Emergent Wetland + Bare Land |
|---|---|---|---|
| Fort Morgan | 0.38 | 0.28 | 0.66 |
| Dauphin Island | 0.00 | 0.67 | 0.67 |
| Petit Bois Island | 0.49 | 0.46 | 0.95 |
| Horn Island | 0.71 | 0.20 | 0.91 |
| Domain | 0.40 | 0.42 | 0.82 |

Petit Bois Island to West Ship Island, which is situated approximately midway between the Mississippi River and Mobile Bay watershed discharges, may be equally suitable habitat for horseshoe crabs. Overall, these data suggest that horseshoe crab presence along the northcentral GOM coast may be mediated by a combination of distance from areas of high freshwater discharge and interannual variation in weather, both of which may affect salinity and water clarity [56].

Within site variation in LCLU may also be important to determining horseshoe crab occurrence. Bare Land and Estuarine Emergent Wetland LCLU classes, where horseshoe crab numbers were highest at all sites, had the greatest area on Petit Bois and Horn Islands. Models including LCLU alone or in combination with site were predictive of live animal, and to some extent carcass, occurrence. These findings suggest that LCLU, particularly the area of Bare and Estuarine Emergent Wetlands, may interact with other environmental attributes to determine horseshoe crab occurrence. Petit Bois and Horn Islands also are within the Gulf Islands National Seashore (GINS) boundary, which may offer some long-term protection against habitat alteration or loss. Dauphin Island and potential habitat along the mainland coast such as the Fort Morgan peninsula are not similarly protected from LCLU changes that may limit areas suitable for horseshoe crabs. The horseshoe crabs found on the Fort Morgan peninsula, for example, were located inside Mobile Bay, where wave and tidal action is reduced, but the area available for spawning is limited by eroding beaches, residential homes, and a combination of emerging peat and remnant trees that fragment the sandy beach. These LCLU attributes likely contributed to the limited occurrence of horseshoe crabs on Fort Morgan. Urbanization and other habitat alterations are known to affect habitat for the American horseshoe crab on the U.S. Atlantic coast [10] and are a major source of habitat loss for Asian horseshoe crabs [57]. While potentially protected from urbanization, the barrier island chains that comprise a major part of horseshoe crab habitat in the northcentral GOM, however, are regularly subjected to intense storms, sea level rise, and dredging activities that increasingly cause land loss [40]. Furthermore, even though the American horseshoe crab is an IUCN red list "vulnerable" species, it is not a priority species for habitat protection and restoration in the GOM or the GINS; hence, habitat area may still be at risk of future loss throughout the region.

The large number of molts found at Petit Bois Island and to a lesser extent Horn Island indicates these sites are important for reproduction and is suggestive that Petit Bois Island could represent a location of peak suitability for spawning habitat in the region. Higher sex ratios were found at the three sites west of the Mobile Bay discharge. Adult sex ratios > 1 in nearshore areas are consistent with known reproductive strategy for horseshoe crabs, where more males than females tend to visit beaches during the spawning season [58]. While no live animals or molts were found on Dauphin Island, the presence of carcasses and the male dominated sex ratio among carcasses may indicate spawning occurred at this site at lower abundance or at times outside our survey period. Crabs in amplexus have been observed on Dauphin Island and molts have been found on the west end of the island in the past (personal observation). The sex ratio closer to 1:1 on Fort Morgan peninsula and the small number of crabs visiting this site could imply the site has limited value in population recruitment [3]. The finding of some live horseshoe crabs (and several pairs in amplexus) at Fort Morgan, however, indicates the need for further assessment of the importance of the southern area of Mobile Bay and similar previously unstudied areas of beach along the mainland of the northcentral GOM coast.

While horseshoe crab size has been found to vary with latitude, the size ranges of all animals measured in this study were consistent with previous studies of horseshoe crabs throughout their range [2, 4, 32, 59, 60]. Molts reflected typical juvenile size classes, and most live animals and carcasses had prosomal widths similar to typical subadult and adult sizes [4, 32, 61]. Size

frequency peaks among molts were comparable to cohorts previously defined based on molts at Petit Bois Island in 2008 and 2009 [4] and indicate these molts represent instars 6 and 9–11 (approximately years 1–4). Size frequency distributions of carcasses and live animals combined suggested these horseshoe crabs were likely instar 17 and age 10 and older; [61–63]. The combination of live horseshoe crabs and carcasses allowed us to better define size demographics of subadult and adult horseshoe crabs in this study. Molts allowed us to further evaluate population characteristics for almost the entire lifespan of horseshoe crabs at our study sites and enhanced our understanding of horseshoe crab occurrence relative to environmental attributes. This approach is transferrable to other habitats with similarly sparse or hard to study populations [4] and the resulting data can support development of formal habitat suitability models in the future.

While live animal studies are ideal for evaluating habitat suitability, these studies are not always feasible, and data from molts and carcasses can be essential to supplement live animal studies. Our data are consistent with a recent study that demonstrated numbers, sex, and size distributions of horseshoe crab molts can enhance population assessment when live animals are not accessible for study [4]. Carcasses of many species from copepods to elephants have been used along with live animal data to infer mortality rates and changes in abundance and populate demographic models [64, 65]. For horseshoe crabs, molts and carcasses increased sample size for understanding population structure but also provided information on habitat use to more fully understand relationships between environmental attributes and horseshoe crab occurrence in this study. In this case, just two years of data substantially increased our knowledge of horseshoe habitat use. Application of this approach, however, depends on the assumption that molts and carcasses represent local live-animal populations. The results of the molt mark-recapture experiment indicated that molts found on local beaches originated from waters near sites and time periods of the surveys in which they were found, with cross-site transport unlikely at distances greater than 4 km in our study area. Carcasses tend to result from animals coming to the beach to spawn and are heavier than molts, making them even less likely than molts to be transported from beaches of origin. Hence, we are confident that our data are locally relevant, and uncertainty related to molt or carcass transport in our study is minimal and outweighed by the consistency of our findings across demographic and environmental metrics. Overall, our data demonstrate that multiple indicators of horseshoe crab presence (live animals, molts, carcasses) can be related to *in situ* and remotely sensed environmental data to inform demographic and habitat assessments, even on short time-scales.

## Conclusions and management implications

Variation in specific attributes of local habitat are important to habitat suitability for horseshoe crabs even among sparse populations on the northern GOM where habitat area does not appear to be limited. Within local habitats, movement to breeding beaches, location of mates, and orientation to food resources depends on fine-scale habitat attributes because sensory cues are effective for horseshoe crabs within short distances, either in the immediate vicinity or up to a few meters away [22, 27, 66]. Previous horseshoe crab surveys have shown large differences in counts among local sites in Florida [67], reduced counts and limited breeding activity associated with local habitat loss on Long Island, New York [68], and that some horseshoe crabs remain in localized areas throughout their lifespan [67–69]. Along with our findings, these studies highlight the importance of local habitat influences.

Ongoing anthropogenic habitat alteration increases the need to understand local impacts on seascape connectivity for horseshoe crab populations. While the importance of connectivity across habitats may vary with distance, the scale and shape of such effects likely need to be

evaluated at local scales for horseshoe crabs as for other species [70]. Studies on coastal areas in the Mediterranean Sea within Marine Protected Areas of community responses to human pressures found that the spatial resolution of threats and the associated community response dictates the nature of management strategies [71]. Broad management strategies are commonly used to estimate threats at a regional scale and are useful to inform conservation from a general perspective, however, local scale information is needed to refine impact evaluations [71]. The need for locally-focused management initiatives has been previously suggested for horseshoe crab populations on Cape Cod in the U.S., due to within-region variation in habitat type and anthropogenic pressures [72]. Hence, this study contributes to a growing body of evidence that conservation efforts for horseshoe crabs will benefit from considering local habitat attributes and the risks to the population from habitat fragmentation and local anthropogenic disturbances.

For the northcentral GOM, our results consistently indicated that the north shoreline of Petit Bois Island and other islands within the GINS are important habitat for horseshoe crabs and merit conservation that will continue to allow shoreline access for horseshoe crabs. In areas like the north shore of the Fort Morgan Peninsula, where habitat may be limited, managers could work with local property owners to preserve existing shoreline and mitigate erosion, while also preserving beach accessibility to horseshoe crabs. Continued monitoring of horseshoe crab habitat in the northcentral GOM is recommended to establish population trends, evaluate changes in shoreline habitat attributes and availability for horseshoe crabs, and support future conservation and management strategies.

## Supporting information

**S1 Table. Kappa statistic (Kappa = observed agreement—expected agreement / 1 –expected agreement) for live horseshoe crabs and LCLU agreement.**
(DOCX)

**S1 Fig. Measuring carcasses, molts (top) and live horseshoe crabs (bottom; amplexed pair) to determine prosomal width.**
(TIF)

## Acknowledgments

Thanks to the Gulf Islands National Seashore for approval to survey on National Park Service Lands, R. Russell at the University of Alabama in Huntsville for editorial assistance with figures and to the DISL staff C. Nelson, J. Delo and E. Hieb; student interns M. Gilroy, E. Lauss, P. Dimens, T. Bilbo,; volunteers S. Carmichael, J. Luvall, and C. Luvall; graduate students A. Aven and E. Condon; and student volunteers T. Mason, L. Gambles, H. Bell, J. Lajoie, K. Marino, and C. Estes for field assistance.

## Author Contributions

**Conceptualization:** Maurice G. Estes, Jr.

**Data curation:** Ruth H. Carmichael.

**Formal analysis:** Maurice G. Estes, Jr.

**Funding acquisition:** Ruth H. Carmichael, Xiongwen Chen.

**Methodology:** Ruth H. Carmichael, Sean C. Carter.

**Project administration:** Xiongwen Chen.

**Resources:** Ruth H. Carmichael, Xiongwen Chen.

**Supervision:** Ruth H. Carmichael, Xiongwen Chen.

**Validation:** Sean C. Carter.

**Writing – original draft:** Maurice G. Estes, Jr.

**Writing – review & editing:** Maurice G. Estes, Jr., Ruth H. Carmichael, Xiongwen Chen.

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
