## [Decision Letter · Decision Letter 0]

11 Aug 2020

PONE-D-20-17862

Environmental factors and occurrence of horseshoe crabs in the northcentral Gulf of Mexico

PLOS ONE

Dear Dr. Estes, Jr.,

Thank you for submitting your manuscript to PLOS ONE. After careful consideration, we feel that it has merit but does not fully meet PLOS ONE’s publication criteria as it currently stands. Therefore, we invite you to submit a revised version of the manuscript that fully addresses all the points raised during the review process. Please consider that one of the reviewers pointed out several manuscript flawsl that shoud be overcome in order to get the manuscript accepted for publication.

We look forward to receiving your revised manuscript.

Kind regards,

João Miguel Dias, Ph.D.

Academic Editor

PLOS ONE

Journal Requirements:

2. In your Methods section, please provide additional location information of the study sites, including geographic coordinates for the data set if available.

4. We note that Figures 1 and 2 in your submission contain map images which may be copyrighted. All PLOS content is published under the Creative Commons Attribution License (CC BY 4.0), which means that the manuscript, images, and Supporting Information files will be freely available online, and any third party is permitted to access, download, copy, distribute, and use these materials in any way, even commercially, with proper attribution. For these reasons, we cannot publish previously copyrighted maps or satellite images created using proprietary data, such as Google software (Google Maps, Street View, and Earth). For more information, see our copyright guidelines: http://journals.plos.org/plosone/s/licenses-and-copyright.

4.1.    You may seek permission from the original copyright holder of Figure s 1 and 2 to publish the content specifically under the CC BY 4.0 license.

4.2.    If you are unable to obtain permission from the original copyright holder to publish these figures under the CC BY 4.0 license or if the copyright holder’s requirements are incompatible with the CC BY 4.0 license, please either i) remove the figure or ii) supply a replacement figure that complies with the CC BY 4.0 license. Please check copyright information on all replacement figures and update the figure caption with source information. If applicable, please specify in the figure caption text when a figure is similar but not identical to the original image and is therefore for illustrative purposes only.

5. We note you have included a table to which you do not refer in the text of your manuscript. Please ensure that you refer to Table 1 in your text; if accepted, production will need this reference to link the reader to the Table.

Reviewers' comments:

Reviewer's Responses to Questions

**Comments to the Author**

1. Is the manuscript technically sound, and do the data support the conclusions?

Reviewer #1: Partly

Reviewer #2: Yes

2. Has the statistical analysis been performed appropriately and rigorously? 

Reviewer #1: I Don't Know

Reviewer #2: Yes

3. Have the authors made all data underlying the findings in their manuscript fully available?

Reviewer #1: Yes

Reviewer #2: Yes

4. Is the manuscript presented in an intelligible fashion and written in standard English?

Reviewer #1: Yes

Reviewer #2: Yes

5. Review Comments to the Author

Reviewer #1: Review for PLOS One: D-20-17862

TITLE: Environmental factors and occurrence of horseshoe crabs in the northcentral Gulf of

Mexico

AUTHORS: Maurice G. Estes, Jr., Ruth H. Carmichael, Xiongwen Chen, Sean Carter

The manuscript describes a study of horseshoe crabs on a set of barrier islands along the northcentral Gulf of Mexico coast toward the western end of the species’ range. As the author’s say, such information and the methods used for collecting such information in low density populations is both valuable in its own right and valuable for what it might say about other low density populations of horseshoe crabs including the at-risk Asian species. Little is known about horseshoe crabs in this region so the data are valuable. Particularly useful elements of this study include an evaluation of the effect of land use/land cover classes on horseshoe crab presence and combining molt, carcass and live animal data to evaluate patterns of presence.

General comments

The results described here are hard to get and certainly it is always important to extract as much as possible from such hard-won data, but some of the analyses and conclusions described in this manuscript seem to be a stretch. First, the islands were visited on 9 occasions in 2012 and 4 occasions in 2013 (based on data in Fig. 4). What is the evidence that these animals are rare? Certainly the population density is not like DE Bay, but how do they compare with other populations in the Gulf of Mexico or other areas such as Maine (at the other end of the species’ range)? Second, while all studies have limitations, this study ignores the fact that the animals were studied from May to September and not during the period when peak numbers of live animals were most likely to be found. Fulford & Haehn 2012 show that on barrier islands in Mississippi Sound including Horn Island (which is one of the islands included in this study) there is a peak of nesting in March and April and Rudloe 1980 (missing from the references) shows clearly that there is a major nesting peak in March and April along the northern Gulf coast of Florida with a second and smaller peak in autumn (Sep-Oct). This means that the authors’ central premise that this is a lower density population than elsewhere in the species’ range is questionable. Also, the authors conclude, “The number of live horseshoe crabs was significantly related to day of the year …” but since they did not study the animals during their peak activity, they are only adding evidence for low activity during summer and a second peak of activity in autumn (Fig. 5). Third, data on size is confounded by not separating males and females for live animals and carcasses (Fig. 6), which means that comparisons with other populations are difficult (data by sex were collected). Fourth, Rudloe 1980 found that water inundation (tide + wind surge) was the most important influence on horseshoe crab numbers in the northeastern Gulf, but the present study does not collect data on actual water levels (which might be available from NOAA). Wind speed and direction are recorded, but not how they influence water height. Fifth, environmental variables such as temperature and salinity are averaged over the study period and used to examine the effects of environmental variables on total observed numbers (“Mean environmental values were computed for each variable for survey years 2012, 2013 and for combined 2012-2013 data for the study area.”) rather than using the data from the date (or week) in which the animals were found. The data on numbers of individuals almost certainly included many zeros on survey dates, which means that a different kind of analysis is needed for data of this type. Sixth, a very interesting addition to the study presented here is the molt transport analysis: 65 tagged molts were placed in the environment at Dauphin Island and 65 at Petit Bois with a reported recapture rate of 2.5% (or about 3 individuals). However, from Fig. 8 (the Fig. numbers do not match the text) it looks as though five tagged molts were recaptured (except that 5 of 130 is 3.9%) so the reader is left wondering about the results of this experiment. It is not clear what the line on Fig. 8 represents. Finally, the strongest results in this study come from the correlations between horseshoe crab abundance and land cover/use categories. Since the environmental variables part of the study revealed little (and may not be analyzed correctly), the authors might consider focusing on these LCLU results. In fact, most of the discussion (lines 518-594) focuses on the parts of this study that are strongest.

Editorial comments

Line 32. The abstract begins with a bit of hyperbole: “This study provides the first regional-scale data…” The study covers a roughly 55 mile stretch along the north-central coast of the Gulf of Mexico with horseshoe crab habitat extending farther to the west (eg. barrier islands of Mississippi) and farther to the east (eg. Escambia County, Florida).

Line 62. “Numbers of horseshoe crabs are reportedly higher to the east along the Florida panhandle…” this would be a good point to reference Rudloe 1980.

Line 77. “… in the northwestern Gulf of Mexico” Except that as you know they are found in microtidal environments.

Line 89. “…where horseshoe crab populations are already sparse and…” the population density in this area is not actually known

Line 105 -114. This paragraph is all about remote sensing. Is this a study using remote sensing?

Line 119 sentence ending, “and most adults thought to migrate from shelf or offshore waters seasonally….” needs a citation.

Line 172. Should be “… discharge in winter and spring than in summer and fall…”

Line 186. Should be “…Surveys were conducted from May to August at biweekly and…”

Lines 189-192. “Thirty-two surveys were completed in 2012 (7 each on Dauphin Island and Fort Morgan Peninsula and 9 each on Petit Bois Island and Horn Island) and 15 surveys were completed in 2013 (4 each on Fort Morgan peninsula, Petit Bois Island and Horn Island, and 3 on Dauphin Island)…” This description does not square with the data given in Fig. 4, which amounts to 9 occasions in 2012 and 4 occasions in 2013. Since not all sites were visited each time, it is not possible to tell on Fig. 4 whether no animals were found or the island was not visited. When exactly did the surveys take place at each site? Fig. 5 presents the same data just for live animals but does not separate the sites. Is effort taken into account in Fig. 5?

Lines 192-194. To continue, the authors state, “…ensuring that a sufficient number of surveys were completed under different environmental conditions to relate horseshoe crab counts to environmental co-variates at each site.” But since many of the surveys are on the same dates, as appears to be the case in Fig. 4 & 5, there are really only 13 different dates or data points to compare with the environmental data. Is this a sufficient sample size? Is this a sufficient sample size to come to conclusions such as horseshoe crab “…presence…[is] mediated by a combination of distance from areas of high freshwater discharge…”

Lines 80-90. Of course a paper can’t cite all relevant publications, but this discussion of the factors influencing horseshoe crab distributions does not reference a number of highly relevant and directly related studies (e.g. Vasquez et al., Cheng et al.)

Line 207. Should be “…Calipers were used for smaller juvenile molts to determine prosomal width…”

Line 247. These are climatic means for Dauphin Island.

Line 252-267. This is not clear. How does this give the height of the ocean floor, relative to what?

Line 289-290. “…to constitute an area large enough that correlations with numbers of live horseshoe crabs, molts or carcasses would not be random.” What is that area? A reference is needed here.

Line 292. Fig. 2. The LCLU classes are a valuable part of this study but this figure is not particularly illuminating. Do all the classes actually occur at the four sites? If some are not included then simplify the multiple colors of the legend. We are told that the horseshoe crabs are on the north sides of the islands but this is not clear from the figure and you can’t see that there is a different LCLU on the north side. The figure legend is not clear. It says “…land cover land use classes resampled from 30 m to 300 m by site for horseshoe crab shoreline habitat analysis with survey data from 2012 and 2013.”

Line 359. Should be “…Islands, and during 2013 99% of all juvenile molts were found on Petit Bois Island…”

Line 362. Table 2. It is not clear what the wind direction and wind speed numbers refer to. They appear to be a mean of the values collected from each island (May-Sep). Is this the number that is used in the analyses (as suggested by the methods, lines 312-313)? If so, then this is a very course way to analyze the data; analyzing the numbers found during each survey (with associated environmental conditions) would seem a more appropriate approach. A southwest wind is said to have an effect on numbers (positive or negative?) but the reader is not told how this relates to the beaches. Which brings up the point that taking means of directions is problematic since 359 is right next to 01. Circular distributions require different stats (not arithmetic mean).

Line 396, 402-404. “… placement, and as of 2019 no additional tagged molts were recovered (Fig 6).” This point is not illustrated with Fig. 6. Maybe Fig. 8?

Line 404. Should be “… counted on beach surveys were correctly associated with the site where they were found.”

Line 407. “Live horseshoe crabs ranged in size from 152 to 280 mm (196 ± 5.01), carcasses ranged in size from 60 to 300 mm (193 ± 2.28 mm),…” Are these adults? Are they males or females? It would be much easier to compare these data with other populations if they were separated into males and females.

Line 415. Are the carcasses adults? A mixture of adults and juveniles? What proportion are adults? Are they male or female?

Line 416. Not sure how these peaks are determined in Fig. 6. Should these be presented as frequencies instead of raw numbers – the legend says frequency but the axis says number?

Line 420. Size data are in Fig. 6 not 7. The adult size data need to be separated by sex. Combining everything does not allow comparisons with other sites.

Line 449 “Fig 8. Daily mean discharge data from the Mobile River into Mobile Bay” these data are given in Fig. 9. Since daily discharge numbers are not related to daily or weekly horseshoe crab numbers, I don’t think this figure is particularly useful. The central point is that the two years differ, which is given in the text with numbers.

Line 464. Should be “The number of live horseshoe crabs was significantly related…”

Line 503. “Accordingly, Petit Bois Island, which is more remote from Mobile Bay…” might be better to say, “Consistent with this view” or some such that is not such a strong conclusion – there are lots of possible reasons for these relationships only some of which were measured in this study. And in fact, the environmental part of this study found few correlations. More comparisons with Fulford & Haehn 2012 and Rudloe 1980 and possibly Brockmann & Johnson 2011 would be useful (size, sex ratios, seasonality).

Line 509. “toward the limit of their distribution in Mississippi.” Should be Louisiana since they are found in Chandeleur Islands.

Line 553. “… has limited reproductive value [3].” Might be better to say limited value in population recruitment (RV is often used in a very different context).

Line 557. “While horseshoe crab size has been found to vary with latitude, the size ranges of all animals measured in this study were consistent with previous studies of horseshoe crabs throughout their range.” Well, adult sizes are consistent with some regions, but how do they compare with areas elsewhere in the Gulf of Mexico or elsewhere in their distribution?

Line 583. “In this case, just two years of data substantially increased our knowledge of horseshoe habitat use where limited or no prior studies had been conducted.” Should be “In this case, just two years of data increased our knowledge of horseshoe crab habitat use…” since Fulford & Haehn was a very fine study.

Line 622. “…anthropogenic pressures [65]. Hence, this study contributes to a growing body of evidence that conservation efforts for horseshoe crabs will benefit from considering local habitat attributes and…” citations to a few of these other studies are needed.

Problems with references (incorrect citations)

8. Smith, D. R., Brockmann, H. J., Beekey, M. A., King, T. L., Millard, M. J. & Zaldivar-Rae, J. 2017. Conservation status of the American horseshoe crab (Limulus polyphemus): a regional assessment. Reviews in Fish Biology and Fisheries, 27, 135-175.

27. Smith, D. R., Brousseau, D. J., Mandt, M. T. & Millard, M. J. 2010. Age and sex specific timing, frequency, and spatial distribution of horseshoe crab spawning in Delaware Bay: Insights from a large-scale radio telemetry array. Current Zoology, 56, 563-574.

50. should be Arthropoda

Missing references

Rudloe, A. 1980. The breeding behavior and patterns of movement of horseshoe crabs, Limulus polyphemus, in the vicinity of breeding beaches in Apalachee Bay, Florida. Estuaries, 3, 177-183.

Cheng, H., Chabot, C. C. & Watson, W. H. 2016. Influence of environmental factors on spawning of the American horseshoe crab (Limulus polyphemus) in the Great Bay estuary, New Hampshire, USA. Estuaries and Coasts, 39, 1142-1153

Vasquez, M. C., Brockmann, H. J. & Julian, D. 2017. Between-population differences in multi-stressor tolerance during embryo development in the American horseshoe crab, Limulus polyphemus. Estuaries and Coasts, 40, 1487-1501.

Reviewer #2: This is a well conducted study...and a well written manuscript. Horseshoe crabs are of such great interest to those who study the ecology of northern Gulf barrier island shorelines. This manuscript provides valuable information which expands current knowledge of barrier island ecology in the northcentral Gulf of Mexico. The research plan was well designed. The field work and data analyses were well executed. This is an interesting study worthy of publication.

6. PLOS authors have the option to publish the peer review history of their article (what does this mean?). If published, this will include your full peer review and any attached files.

Reviewer #1: No

Reviewer #2: No

---

## [Author Response · Author response to Decision Letter 0]

22 Oct 2020

Response to editorial and reviewers’ comments to the Author

Reviewer #1

1. What is the evidence that these animals are rare? 

We do not claim horseshoe crabs are rare. The abundance of horseshoe crabs in the northern Gulf of Mexico is unknown. We do know, based on our own observations and the existing literature, that populations are less dense (sparse) compared to populations in the eastern Gulf of Mexico and on the U.S. Atlantic coast, as discussed in the introduction. We took care to use the term ‘sparse’, rather than ‘rare’ throughout the text for this reason. In this revised version, we additionally replaced the term ‘sparse’ with ‘less densely populated’ in some parts of the text for clarity and we confirmed that the term ‘rare’ is not in the text. 

2. Surveys conducted May – Sep, do not include the previously estimated period of peak of spawning during late April and May. Hence, populations may not be sparse.

While the reviewer is correct that our surveys did not cover a portion of the spawning period that may include a second (or in this case, a first peak), it is important to note that this study is not a spawning survey and not intended to focus primarily on spawning habitat (many of this reviewers comments seem to derive from a spawning-focused lens). The difference in horseshoe crab density at the northcentral Gulf of Mexico sites we studied (including the data from Fulford and Haehn noted by the reviewer) indicate the density of horseshoe crabs is so much lower at these sites than at sites in peninsular Florida and the mid-Atlantic, that inclusion of these time periods could not possibly change the description of these populations from ‘sparse’ in comparison. The Rudloe data cited by the reviewer make this same point, and these data were collected from peninsular Florida, which has a higher density of crabs. 

3. Data on size is confounded by not separating males and females for live animals and carcasses. 

This is a good point, and we have included these values in the text to show the data separated by sex. Thank you for this suggestion.

4. Rudloe 1980 found that water inundation (tide + wind surge) was the most important influence on horseshoe crab numbers in the northeastern Gulf, but the present study does not collect data on actual water levels. 

Rudloe 1980 is one of the foundational spawning population studies in peninsular Florida. That work was done to define conditions of spawning. Since that time the relationship between numbers of spawning crabs and water height has been very well documented, and sampling protocols have been developed to normalize for this variable. Although we were not targeting spawning animals, to maximize likelihood of finding animals at survey sites, we used these established protocols (sampling within 2 hours of high tide) to normalize for water height. For this reason, we could focus on other factors that varied among sites as is now common practice. 

5. Averaging by month or year does not consider zeros in the data; the data on numbers of individuals almost certainly included many zeros …, which means that a different kind of analysis is needed for data of this type. 

The reviewer is correct that there are zeros in the dataset as can be seen in the table and figures, but we made certain that all statistical tests we applied were suitable to our dataset prior to analyses, and we did not violate any assumptions of those tests. 

6. The molt transport analysis may have incorrectly calculated the % recovery.

The % recovery is accurately reported as 2.5%. Five molts were recovered out of 195 marked and placed in the field (not 130 as assumed by the reviewer); molts were placed 3 times at n = 65 each time. Thank you for checking on this calculation.

7. The LCLU is the strongest part of the paper and should be emphasized.

We thank the reviewer for the kind words and agree that this portion of the study is very important. The preceding components in the manuscript are needed to provide context for these analyses. To highlight the importance of this work, however, we gave this aspect of the study a priority position in the discussion.

8. Additional comments

a. Line 32. The abstract begins with a bit of hyperbole: “This study provides the first regional-scale data…” It is not the first because Fulford and Haehn have data for this region.

Fulford and Haehn data cover only spawning animals on two islands in Mississippi. This study spans a much larger spatial area, roughly 55 km along MS and AL coasts of the northcentral Gulf of Mexico. The Fulford and Haehn study also did not collect comprehensive environmental data nor make comparisons to the timing and location of the animals, limiting the application of those data to understand habitat suitability without collection and analyses of additional data. Hence, this study is the first to make this effort. We do not, however, think this is a major point and have removed ‘first’ from the abstract.

b. Line 62. “Numbers of horseshoe crabs are reportedly higher to the east along the Florida panhandle…” this would be a good point to reference Rudloe 1980.

Reference added

c. Line 77. “… in the northwestern Gulf of Mexico”, except that as you know they are found in microtidal environments.

While horseshoe crabs are found in microtidal environments within range of apex populations, their numbers decline moving into these habitats. For this reason and other evidence associated with evolutionary-scale changes in geomorphology, the lack of tidal magnitude is the prevailing theory for modern horseshoe crabs not being found in the northwestern Gulf of Mexico (refer to citations in the text and several easily discoverable publications by the late Dr. Carl Shuster and/or Dr. Robert Loveland). Note, we also included the statement that this idea “may explain” the distribution because this idea is a theory.

d. Line 89. “…where horseshoe crab populations are already sparse and…” the population density in this area is not actually known

See comments above under #1 and #2.

e. Line 105 -114. This paragraph is all about remote sensing. Is this a study using remote sensing?

Yes, the study uses remotely sensed data for the land cover land use analysis (C-CAP Landsat derived) and NLDAS data for precipitation, wind speed and wind direction as noted in the methods. This point is an important component of the manuscript and is also addressed in the discussion because of the potential benefit these data provide to supplement and enhance in situ data collection.

f. Line 119 sentence ending, “and most adults thought to migrate from shelf or offshore waters seasonally….” needs a citation.

Reference added: Shuster and Botton (1985), A contribution to the population biology of horseshoe crabs Limulus polyphemus in Delaware Bay. Estuaries 8:363–372

g. Line 172. Should be “… discharge in winter and spring than in summer and fall…”

Thank you for catching this type-error; “is” replaced with “in”.

h. Line 186. Should be “…Surveys were conducted from May to August at biweekly and…”

“typically” deleted.

i. Lines 189-192. When exactly did the surveys take place at each site? Fig. 5 presents the same data just for live animals, but does not separate the sites; Is effort taken into account?

This figure has been edited to clarify numbers at each site, including sites not surveyed for a given date (now indicated by an X in new figure 4). On some dates surveys were missed due to weather conditions, especially late in the summer of 2012 when one hurricane and numerous thunderstorms made small boat travel unsafe. The pattern of occurrence (shown in Fig. 3) did not change when normalized to sampling effort among sites. Hence, level of effort is not the driver of the observed data differences. To clarify this point, the level of effort (occurrence per day) was added to Fig. 3. 

j. Lines 192-194. …there are 13 different dates or data points to compare with the environmental data. Is this a sufficient sample size to come to conclusions such as horseshoe crab “…presence…[is] mediated by a combination of distance from areas of high freshwater discharge…”

There were a sufficient number of samples to detect patterns in the data and perform statistical analyses that yielded valid outcomes. While a larger sample size is always desirable, we had sufficient sampling during the varying weather conditions at the 4 sites to allow comparisons. Given the variation in environmental conditions and among sites, including variation in discharge, our reported findings on horseshoe crab occurrence and relationships to environmental attributes are reasonably established. As noted by the reviewer, these data are hard won and will always have some limitations. The text has been edited throughout out to clarify that our results are based on the period of study and with an eye to clarifying the potential limitations of the interpretation.

k. Lines 80-90. Of course a paper can’t cite all relevant publications, but this discussion of the factors influencing horseshoe crab distributions does not reference a number of highly relevant and directly related studies (e.g. Vasquez et al., Cheng et al.)

The suggested references have been added.

l. Line 207. Should be “…Calipers were used for smaller juvenile molts to determine prosomal width…”

“molts” added to statement as noted.

m. Line 247. These are climatic means for Dauphin Island.

The reviewer is correct, these values are climatic means for Dauphin Island, which are provided as a proxy for climate conditions in the study area. Comparable climate data were not available for the other 3 sites surveyed. For clarity, this table was moved to the results section and the comparison was integrated into the results section. 

n. Line 252-267. How does this give the height of the ocean floor, relative to what?

A DEM is a 3D representation of land or subsurface elevation values to represent terrain or bathymetric with the dataset referenced. Acoustic techniques such as echo sounders are often used to obtain points or values in marine regions. The Vdatum bathymetry values can to be related to various tidal datums. The bathymetry values used in this study are related to mean high water.

o. Line 289-290. “…to constitute an area large enough that correlations with numbers of live horseshoe crabs, molts or carcasses would not be random.” What is that area? A reference is needed here.

The C-CAP land cover land use data (LCLU) is at 30 m native resolution (30 m x 30 m grid), which was only a 30 m shoreline length and very fragmented with other lclu classes. These data were resampled to coarser resolutions of 100 m and 300 m to identify clusters of LCLU classes that could reasonable represent major shoreline habitat characteristics. 300 m was selected as noted in the manuscript (300 m x 300 m grid), which represents a shoreline length of about 300 m per LCLU class. Reference added for the C-CAP data. 

p. Line 292. Fig. 2. The LCLU classes are a valuable part of this study... Do all the classes occur at the four sites? The figure legend is not clear. 

The figure was edited to make the shorelines more visible and clarify the caption relative to horseshoe crab habitat. Not all of the classes occur on the four sites. Given the narrow width of some of the sites (Dauphin and Petit Bois Islands) the same pixel and class may be in one grid cell that covers both north and south shorelines. 

q. Line 359. Should be “…Islands, and during 2013 99% of all juvenile molts were found on Petit Bois Island…”

We reworded the sentence to maintain the comma after 2013, which is grammatically necessary. 

r. Line 362. Table 2. It is not clear what the wind direction and wind speed numbers refer to. … taking means of directions is problematic since 359 is right next to 01. Circular distributions require different stats (not arithmetic mean).

We thank the reviewer for pointing out the need to clarify the methods used to calculate wind direction. The methods have been revised in the text. Wind speed was considered as the force of the wind during the time surveys were conducted for that site. Wind direction is the direction (0-360 degrees) from which the wind is blowing (e.g. 180° would represent winds traveling from the south to the north). The reviewer is correct that we needed to (and did) calculate the mean of circular quantities. Briefly, the means of the cosines and sines of each angle (wind direction during a survey) were computed. The sum of the cosine and sines was divided by the number of surveys (n). The ratios of the sum of the sines to sum of the cosines was computed and the arctangent of this value was considered the mean angle of the wind direction in degrees. 

s. Line 396, 402-404. “… placement, and as of 2019 no additional tagged molts were recovered (Fig 6).” This point is not illustrated with Fig. 6. Maybe Fig. 8?

Our apologies. In the previous version of the text some figures were mis-labeled. We thank the reviewer for pointing out this error, which has been fixed in the revised text. 

t. Line 404. Should be “… counted on beach surveys were correctly associated with the site where they were found.”

Statement edited to add “where they were” as noted.

u. Line 407; line 415. It would be much easier to compare these data with other populations if they were separated into males and females.

See response to #3 above.

v. Line 416. Not sure how these peaks are determined in Fig. 6. Should these be presented as frequencies instead of raw numbers – the legend says frequency but the axis says number?

The figure is a histogram showing the frequency distribution of horseshoe crabs at sizes depicted in the bins on the x-axis. Mixed distribution analyses were previously performed on these size classes as described in Estes et al. (2015) and cited in the text; this study compares newly collected data to those established size distributions.

w. Line 420. Size data are in Fig. 6 not 7. The adult size data need to be separated by sex. 

See comments to #3 and #8s above.

x. Line 449; Fig. 9., Since daily discharge numbers are not related to daily or weekly horseshoe crab numbers, I don’t think this figure is particularly useful. The central point is that the two years differ, which is given in the text with numbers.

Additional comparisons on horseshoe crab occurrence, size and sex ratios are in the revised text. The discharge data are directly related to the years and time periods that were surveyed. Without the figure, the pattern of discharge fluctuations when surveys were conducted could not be visualized. Also, the figure highlights (along with the text numbers) the magnitude and temporal fluctuations of discharge across the two years of study.

y. Line 464. Should be “The number of live horseshoe crabs was significantly related…”

“were” changed to “was” 

z. Line 503. “Accordingly, Petit Bois Island, which is more remote from Mobile Bay…” might be better to say, “Consistent with this view”.

Text was changed as suggested.

aa. Line 509. “toward the limit of their distribution in Mississippi.” Should be Louisiana since they are found in Chandeleur Islands.

“Mississippi” changed to “Louisiana”

bb. Line 553. “… has limited reproductive value [3].” Might be better to say limited value in population recruitment.

“reproductive value” deleted and “population recruitment” added.

cc. Line 557. … adult sizes are consistent with some regions, but how do they compare with areas elsewhere in the Gulf of Mexico or elsewhere in their distribution?

The sizes are comparable with other horseshoe crabs found in MS and LA barrier islands (Fulford and Haehn) as well as the northern and western Gulf coast of Florida. Text was added to make these demographic comparisons. It is noted, however, that Fulford and Haehn and Rudloe targeted spawning animals (we did not) and in some cases reported data in ways that make direct comparison difficult.

dd. Line 583. Should be “In this case, just two years of data increased our knowledge of horseshoe crab habitat use…” since Fulford & Haehn was a very fine study.

Text was edited to delete “where limited or no prior studies had been conducted”. While the Fulford and Haehn study is complementary to this work, it is a very different study in scope; limited in spatial scale and to spawning animals only, without regard to comprehensive environmental conditions.

ee. Line 622. …this study contributes to a growing body of evidence that conservation efforts for horseshoe crabs will benefit from considering local habitat attributes …” citations to a few of these other studies are needed.

This sentence is the concluding sentence to an entire paragraph dedicated to this point; note all of the citations throughout the paragraph. In addition, we added Botton et al. (2018), which also specifies geomorphology on a local scale is detectable and can mediate horseshoe crab occurrence.

9. Problems with references (incorrect citations)

Citations corrected

10. Missing references: Rudloe, A. 1980.; Cheng et al. 2016; Vasquez et al. 2017

References added

Reviewer #2

This is a well conducted study...and a well written manuscript. Horseshoe crabs are of such great interest to those who study the ecology of northern Gulf barrier island shorelines. This manuscript provides valuable information which expands current knowledge of barrier island ecology in the northcentral Gulf of Mexico. The research plan was well designed. The field work and data analyses were well executed. This is an interesting study worthy of publication.

We thank this reviewer for the kind words and recognition of the value of the dataset.

Additional authors’ revisions: In addition to the suggested revisions by the reviewers, we copyedited the manuscript to correct some minor type errors (spelling, extra spaces, etc.), consistently referred to the region as northcentral GOM (rather than nGOM) for clarity, and made some modifications to abbreviations and rounding in the Tables for consistency and clarity. We also updated the figure numbers and in-text references to the figures and tables. These minor changes are not tracked in the text to avoid confusion with reviewer suggestions.

---

## [Editor Report · Decision Letter 1]

23 Nov 2020

Environmental factors and occurrence of horseshoe crabs in the northcentral Gulf of Mexico

PONE-D-20-17862R1

Dear Dr. Estes, Jr.,

We’re pleased to inform you that your manuscript has been judged scientifically suitable for publication and will be formally accepted for publication once it meets all outstanding technical requirements.

Kind regards,

João Miguel Dias, Ph.D.

Academic Editor

PLOS ONE
---

## [Editor Report · Acceptance letter]

11 Dec 2020

PONE-D-20-17862R1 

Environmental factors and occurrence of horseshoe crabs in the northcentral Gulf of Mexico 

Dear Dr. Estes, Jr.:

I'm pleased to inform you that your manuscript has been deemed suitable for publication in PLOS ONE. Congratulations! Your manuscript is now with our production department. 

Kind regards, 

on behalf of

Dr. João Miguel Dias 

Academic Editor

PLOS ONE